# Comparison of Antibody Responses against Two Molecules from *Ascaris lumbricoides*: The Allergen Asc l 5 and the Immunomodulatory Protein Al-CPI

**DOI:** 10.3390/biology12101340

**Published:** 2023-10-16

**Authors:** Velky Ahumada, Josefina Zakzuk, Lorenz Aglas, Sandra Coronado, Peter Briza, Ronald Regino, Fátima Ferreira, Luis Caraballo

**Affiliations:** 1Institute for Immunological Research, University of Cartagena, Cartagena de Indias 130012, Colombia; vahumadac@unicartagena.edu.co (V.A.); jzakzuks@unicartagena.edu.co (J.Z.); scoronador@unicartagena.edu.co (S.C.); rreginol@unicartagena.edu.co (R.R.); 2Department of Biosciences, University of Salzburg, 5020 Salzburg, Austria; lorenz.aglas@sbg.ac.at (L.A.); peter.briza@plus.ac.at (P.B.); fatima.ferreira-briza@plus.ac.at (F.F.)

**Keywords:** IgE, allergen, parasites, helminths, allergenicity

## Abstract

**Simple Summary:**

Helminth infections may have different effects on human health, including risk or protection from other diseases. Ascariasis (caused by *Ascaris lumbricoides*), the most common soil-transmitted helminthiasis, can induce allergic responses, and also immunosuppression. During ascariasis, antibodies for many *A. lumbricoides* antigens are produced; however, there is no clear information about the concurrent IgE, IgG4 and IgG production as well as their influences on the actual allergic reactions. In this study, we evaluated antibody responses to two *A. lumbricoides* molecules, rAsc l 5 and rAl-CPI (an anti-inflammatory product), in an *A. lumbricoides* endemic population and explored their relationship with the infection and asthma. Our results show that both molecules induce specific antibodies, but, in contrast to rAl-CPI, rAsc l 5 activates cells associated with allergic reactions in some individuals. All together, these data suggest that these molecules have differences in the elicited immune response.

**Abstract:**

Immunity to *Ascaris lumbricoides* influences the pathogenesis of allergic diseases. Antibody responses to its proteins have been found to be associated with asthma presentation; however, helminth products that induce immunosuppression have been reported, which also raise specific antibodies. We aimed to evaluate antibody responses (IgE, IgG4 and IgG) to two *A. lumbricoides* molecules, Asc l 5 and Al-CPI (an anti-inflammatory Cysteine Protease Inhibitor), in an endemic population, exploring their relationships with the infection and asthma. The two molecules were produced as recombinant proteins in *E. coli* expression systems. Specific antibodies were detected by ELISA. Lower human IgE, but higher IgG4 and IgG antibody levels were observed for Al-CPI than for rAsc l 5. The IgE/IgG4 isotype ratio was significantly higher for Asc l 5 than for Al-CPI. In humans Al-CPI did not induce basophil activation as has been previously described for Asc l 5. In mice, Al-CPI induced fewer IgE responses, but more IgG2a antibody titers than rAsc l 5. Our results suggest that these molecules elicit different patterns of immune response to *A. lumbricoides*.

## 1. Introduction

*Ascaris lumbricoides* is a soil transmitted parasite that causes ascariasis, one of the most prevalent helminthiases in humans [1]. Its success in infecting susceptible people may be explained by its ability to cause immunosuppression of the host’s immune responses [2], mainly in places with limited sanitation and poor hygiene practices. For example, it has been found, that in certain populations where helminth infections are prevalent, a positive skin prick test for environmental allergens, such as house dust mites, (HDM) shows that they are reduced [3,4], which could be explained by IL-10 production [5] and the effects of immunomodulatory components [6], among them cystatins of the family of cysteine pro-tease inhibitors (CPIs) [7,8,9,10] that interfere with antigen processing via MHC class II molecules [11,12,13,14]. It has been shown that *A. lumbricoides* CPI (Al-CPI) ameliorates symptoms of inflammation in mice models of both dextran sodium sulphate-induced colitis and allergic airway inflammation [7,8]. Thus, Al-CPI might be developed as an anti-inflammatory candidate [15]. However, it is well known that helminths induce specific IgE responses to several of their molecules [16,17], sometimes leading to allergic reactions that would impede their use in humans. This makes the evaluation of the allergenic capacity of *A. lumbricoides* molecules proposed as therapeutic tools mandatory. Asc l 5 is the fourth allergen identified from *A. lumbricoides* [18]. It is a member of the SXP/RAL-2 protein family which includes proteins from various nematode species [19]. The IgE response to Asc l 5 was about 52% in the sample study, but specific IgG antibodies were not investigated. Actually, there are few reports about the human IgG antibody responses to purified *A. lumbricoides* molecules [20,21]. As such, the aim of this work was to identify and compare antibody responses to the recombinant molecules rAl-CPI (an immunomodulator) and rAsc l 5 (an allergen) in a population from the Caribbean coast of Colombia, where ascariasis is endemic [22]. In addition, we compared the specific antibody responses induced by both molecules in a mouse model of passive cutaneous anaphylaxis (PCA).

## 2. Materials and Methods

### 2.1. Production of Recombinant Proteins

Asc l 5 (Asc l 5, GenBank Accession Number MN275230) was cloned into a pET-45b+ (Genscript, Piscataway, NJ, USA) vector and expressed as 6xHis-tagged protein in *E. coli* Origami (DE3). Al-CPI (GenBank Accession Number HQ404231.1) was cloned into a pQE30 vector and expressed as an 6xHis-tagged protein in *E. coli* SG13009. Bacterial cultures were grown in Luria–Bertani medium under conditions described previously [7,9]. Recombinant versions of these proteins are named as rAsc l 5 and rAl-CPI, respectively.

### 2.2. Peptide Analysis by Nano-Liquid Chromatography–Tandem Mass Spectrometry (LC-MS/MS)

*A. lumbricoides* extract (1000 µg/mL) was digested using the ProteoExtract All-in-One Trypsin Digestion Kit (EMD Millipore, Billerica, MA, USA) and desalted using C18ZipTips (EMD Millipore, Billerica, MA, USA). Resulting peptides were separated by reverse-phase nano-high performance liquid chromatography (HPLC, Dionex Ultimate 3000, Thermo Fisher Scientific, Bremen, Germany, column: PepSwift Monolithic Nano Column, 100 µM × 25 cm, Dionex). The column was developed with an acetonitrile gradient (Solvent A: 0.1% (*v*/*v*) FA/0.01% (*v*/*v*) TFA/5% (*v*/*v*) ACN; solvent B: 0.1% (*v*/*v*) FA/0.01% (*v*/*v*) TFA/90% (*v*/*v*) ACN; 5–45% B in 60 min) at a flow rate of 1 µL/min at 55 °C. The HPLC was directly coupled via nano electrospray to a Q Exactive Orbitrap mass spectrometer (Thermo Fisher Scientific). Capillary voltage was 2 kV. For peptide assignment, a top 12 MS/MS method was used with the normalized fragmentation energy set to 27%. The protein was identified with PEAKS Studio 8 (Bioinformatics Solutions, Waterloo, ON, Canada) using the UniProt (SwissProt/TrEMBL) sequence database. Only peptides with high confidence scores (−10lgp ≥ 35, corresponding to false discovery rate (FDR) < 0.5%) were considered in the database searches [18]. Abundance of Al-CPI in the extract was calculated based on the “Area” parameter from the summarized proteomic search results [23]. 

### 2.3. Circular Dichroism (CD) Spectroscopy

CD spectra were recorded using a JASCO J-815 spectropolarimeter fitted with a PTC-423S Peltier type single position cell holder (Jasco, Tokyo, Japan), in 10 mM K2HPO4/KH2PO4. Samples were measured from 190 to 260 nm (CD far–UV spectrum) at resolution of 1 nm, with 1 nm bandwidth and a scanning speed of 1 nm/s. Five spectra were averaged and the background of the buffer solution was subtracted. Data were presented as mean residue molar ellipticity. Recombinants (concentration of 100 µg/mL) were evaluated at 20 and 95 °C, respectively. For calculation of denaturation temperature (Tm), the protein samples were subjected to gradual increases in the temperature from 20 to 95 °C. CD thermal denaturation curves were established. The thermal denaturation of the proteins was evaluated from 20 to 95 °C at 222 nm wavelength and a temperature ramp rate of 1 °C/min [24]. Excitation wavelength was 280 nm. The mean residue ellipticities (θ) were calculated as described previously [25,26].

### 2.4. Serum Samples and Stool Examination by Kato-Katz

Seropositivity to *A. lumbricoides* antigens was estimated in a rural population from the northern coast of Colombia (Santa Catalina, Bolívar) [18,22] with a high burden of ascariasis. Most of the participants included in this study (*N* = 298, aged 1 to 88 years) had available serum samples for ELISA and data for parasitological variables (number of available samples from the participants are indicated in the results as needed). A physician interviewed participants for asthma diagnosis [22]. *A. lumbricoides* infection was diagnosed by parasitological analysis by counting eggs per gram of feces (epg) [26]. 

### 2.5. Evaluation of Antibody Responses (Specific IgE, IgG4 and IgG) by Enzyme-Linked Immunosorbent Assay (ELISA)

IgE was determined as described previously [18]. For the determination of specific IgG4 and IgG, serum samples and conjugate dilutions were obtained by titration. Optical density (OD) values were detected in duplicate using ELISA. Each well was coated with 1 µg of antigen diluted in sodium carbonate/bicarbonate buffer. Microtiter plates (IMMULON 4HBX, Thermo Fisher Scientific) were incubated overnight at 4 °C in humid chambers, washed five times (PBS pH 7.4, 0.1% Tween 20) and blocked with 1% bovine serum albumin. The plates were washed again, and serum samples (diluted 1:5 for IgE; 1:25 for IgG4 or 1:50 for IgG) were added and incubated overnight at room temperature in a humid chamber, then washed and incubated with the conjugates, respectively, as follow: 2 h with anti-human IgE–alkaline phosphatase conjugate (Sigma, St. Louis, MO, USA) diluted 1:500 or anti-human IgG4–alkaline phosphatase conjugate (1:2000). Anti-human IgG–alkaline phosphatase conjugate (1:30,000) was used for incubation for 1 h. For the development of the reaction p-nitrophenyl phosphate substrate (1 mg/mL; Sigma), diluted in 10% diethanolamine, 0.5 mM MgCl_2_ was used. The reaction was stopped with 100 µL of 3N NaOH, and absorbance was measured at 405 nm in a spectrophotometer (Spectra Max 250; Molecular Devices, Sunnyvale, CA, USA). Positive and negative control sera were used in each experiment and PBS was a control for non-specific binding of the anti-human IgE, IgG4 or IgG–alkaline phosphatase conjugates. Cut-off values to define positive or negative antibody responses to Asc l 5 or Al-CPI were calculated as the mean optical density (OD) of 9 negative controls + 3 SD for IgE or IgG4 and as the mean optical density (OD) of 7 negative controls + 3 SD for IgG. Differences in the number of controls used for calculating the cut-offs depended on the availability of negative serum samples. Seropositivity was defined as a positive specific IgE, IgG4 or IgG result detected by ELISA to any recombinant antigen. The cut-off values were: For IgE to rAsc l 5: 0.145; rAl-CPI: 0.141. For IgG4 to rAsc l 5: 0.170; rAl-CPI: 0.170. For IgG to rAsc l 5: 0.330; rAl-CPI: 0.316. Seropositivity was defined as a positive specific IgE, IgG4 or IgG result detected by ELISA to any recombinant antigen. IgE/IgG and IgE/IgG4 ratios were calculated from the end-point OD values for the three isotypes using IgE as the numerator. 

Human IgE detection by ELISA was performed before and after IgG depletion to evaluate the blocking activity of this last isotype on IgE binding. For IgG depletion, sera were incubated with protein G Sepharose (Bio Vision, Milpitas, CA, USA) for 4 h at room temperature. After centrifugation, protein G Sepharose was added to the supernatant and again incubated. This procedure was repeated four times [27].

### 2.6. Evaluation of Allergenicity

The CD203c-based basophil activation assay was used for ex vivo evaluation of the allergenic activity of recombinant antigens as described elsewhere [18]. Briefly, fresh blood was drawn from five patients sensitized to Al-CPI and Asc l 5 (i.e., a positive IgE result in ELISA) who also had a diagnosis of asthma and a negative control participant who was not sensitized to either to *A. lumbricoides* or purified antigens, and then cells were stimulated with *Ascaris* extract (10 µg/mL), rAl-CPI or rAsc l 5 at different concentrations (10, 1 and 0.1 µg/mL). Mean fluorescence intensity (MFI) results for CD203c in gated basophils were analyzed as stimulation indexes (SI), considering a value ≥2.0 as a positive result [18]. For basophil histamine release, heparinized blood was drawn from 8 subjects sensitized to *A. lumbricoides* extract and also to rAl-CPI. Histamine release was induced by incubating 200 µL of fresh blood with 200 µL of serial concentrations (0.0001, 0.001, 0.01 and 0.1 µg) of rAl-CPI or *A. lumbricoides* extract, allergen dilution buffer and anti-human IgE (Histamine Release kit; IBL International GmbH, Hamburg, Germany). Histamine concentrations were quantified by ELISA (IBL International) following manufacturer instructions. Total histamine release was determined in cells treated with hypotonic solution and regarded as 100% release.

### 2.7. Production of Antibodies to Recombinant Asc l 5 or Al-CPI, Immunization Schedule and Determination of Titers by ELISA

Antisera against recombinant rAsc l 5 and rAl-CPI antigens were raised in 6- to 8-week-old female BALB/c mice (Instituto Nacional de Salud, INS, Bogotá, Colombia). Mice received intraperitoneal injections of the recombinant proteins (20 µg) suspended in 2 mg of Alum (Imject, Thermo Scientific, Waltham, MA, USA) three times in a seven-day interval, and blood was sampled 7 days after the last injection. Mice were kept in a specific pathogen-free environment (22 °C, 50–60% humidity, and 12-h light/dark cycle) and fed with a standard pellet diet and drinking water ad libitum [7]. Ovalbumin (OVA group) was the positive control of the experiment and PBS-treated mice were the negative control (PBS group). 

For specific immunoglobulin determinations, MaxisorpTM microtiter plates were coated with 0.5 µg/well of rAsc l 5, rAl-CPI or OVA, respectively; followed by overnight incubation at 4 °C and washed 4 times with Tween 20 0.1% PBS. Wells were then blocked with PBS 1% BSA 0.05% Tween 20 for 3 h at room temperature. Plates were washed (5X) and incubated overnight with diluted mouse plasma samples at 4 °C (see Appendix A for plasma dilution in each assay). After 5 washes, wells were incubated for 1 h at RT with biotin labeled anti-mouse IgE, IgG1 or IgG2a (see below dilution of samples and conjugates). After 5 washes, ExtrAvidin alkaline-phosphatase (dilution 1:4000, Sigma-Aldrich, St. Louis, MO, USA) was added and incubated for 1 additional hour. P-nitrophenyl phosphate (1 mg/mL; Sigma) was used as the substrate solution (The incubation time for the reaction development after adding the substrate was 30 min for IgG1 and IgG2a or 1 h for IgE, respectively, for each protocol of antibody determination). Optical densities were read at 405 nm in a spectrophotometer. To increase the sensitivity of the IgE ELISA, IgG was depleted from plasma by incubation with protein G Sepharose [28]. Dilutions of plasma samples for detection of antibodies and conjugate dilutions were obtained by titration as follow: IgE plasma samples (1:10) and conjugate biotin rat anti-mouse IgE (1:1000, eBioscience, San Diego, CA, USA). IgG1 plasma samples (1:80,000) and conjugate biotin rat anti-mouse IgG1 (1:10,000, eBioscience, San Diego, CA, USA). IgG2a plasma samples (1:320) and conjugate biotin rat anti-mouse IgG2a (1:1000, eBioscience, San Diego, CA, USA). 

### 2.8. Western Blot

Mouse IgG from rAl-CPI sensitized mice and PBS-injected mice (mock) were diluted 1:1000 in blocking buffer and incubated for one hour at RT with A. lumbricoides extract or rAl-CPI, and transferred to PVDF membranes after SDS-PAGE separation. Strips were rinsed twice and washed three times with 0.1% Tween 20 PBS. Horseradish peroxidase conjugated anti mouse-IgG (diluted 1:100,000 in blocking buffer) was used as a secondary antibody and incubated for 1 h at RT. The membrane was rinsed twice and washed three times with 0.1% Tween 20 PBS. Chemiluminiscence reaction was developed by the addition of a 1:1 mixture of the SuperSignal West Femto Maximum Sensitivity Substrate™ and incubation for 5 min in the dark. Images were obtained at different times in a specialized chemilumiscence image detector (G: Box, Syngene, UK). 

### 2.9. Passive Cutaneous Anaphylaxis

A passive cutaneous anaphylaxis (PCA) model was performed as described previously for rAsc l 5 [18]. Dye extravasation was quantified using an Evans Blue concentration curve and measuring samples at 620 nm in a spectrophotometer (Multiskan™ Thermo Fisher Scientific, Waltham, MA, USA).

### 2.10. Statistical Analysis

Analyses were conducted using SPSS version 20.0 (SPSS, Chicago, IL, USA) and GRAPHPAD PRISM version 5.01 for Windows (GraphPad Software, San Diego, CA, USA). IgE, IgG4 and IgG values of individual participants were not normally distributed; therefore, they were reported as a geometric mean and 95% confidence interval (CI). The Mann–Whitney U-test or Kruskal–Wallis test was used for comparison of continuous variables as needed. Differences between proportions of subjects with positive antibody responses (seropositive) were analyzed by a Pearson chi-square test (Taking into account the cut-off determined by ELISA). For all analyses, *p* values < 0.05 (two tailed) were considered significant. In the analysis of mouse antibodies, for the comparison of the means of the two groups, the student’s *t*-test was used; for more than two groups, one-way ANOVA and Dunnett’s multiple comparison test were used to compare the means. In these cases, mean ± standard error of the mean (SEM) was used for each group. Correlograms were obtained in R, using the *corrplot* package. A Spearman correlation test was used due to the non-normal distribution of most variables.

## 3. Results

### 3.1. Circular Dichroism Reveals Recombinant Asc l 5 and Al-CPI Are Well-Folded Products That Differ in Their Thermostability

The secondary structure determination of rAl-CPI was consistent with the expected theoretical results for its primary sequence and solved three-dimensional structure [9]. CD spectra of rAl-CPI at 20 °C and 95 °C were similar (Figure 1A). In contrast, rAsc l 5 was denatured by heating (Figure 1B). The denaturation temperature was Tm = 58.2 ± 0.37 °C. 

We also assessed, using Western Blot, if anti-Al-CPI antibodies raised in immunized mice were able to recognize the natural antigen in the A. lumbricoides extract, finding two bands in the 10–13 kDa range, which is in accordance with the expected size for Al-CPI (Appendix A). Al-CPI content in the extract was also confirmed by tandem mass spectrometry (MS/MS) sequencing. High sequence coverage (85.7%) of the mature protein (Al-CPI GenBank accession number: HQ404231.1) was obtained (Appendix A). Based on proteomic analysis of the extract, Al-CPI was the fifth most abundant protein in the preparation; in contrast, Asc l 5 had lower representativeness in the extract.

### 3.2. Lower IgE Levels but Higher IgG/IgG4 Response Was Found for Al-CPI Than for Asc l 5

As shown in Table 1, the number of subjects with IgG4+ responses to Al-CPI was twice as high as the frequency rate obtained for the Al-CPI specific IgE response (*p* < 0.0001), while the frequency of the IgG4+ cases to Asc l 5 was similar to the IgE response (68.6 vs. 52.2%, *p* = 0.643). 

There were also differences in the antibody response in regard to sex. Females showed significantly greater median O.D. values for specific IgE determinations to Asc l 5 and Al-CPI, respectively. This difference was also observed in the number of cases with a positive IgE test for Al-CPI (female: 44.6% vs. male: 30.6%, χ^2^: 4.9, *p* = 0.02) but not for Asc l 5 (females: 54% vs. males: 47% χ^2^: 1.2, *p* = 0.28). Sex was not associated with IgG and IgG4 responses.

Median O.D. values for sIgE determinations to Asc l 5 were greater than those against Al-CPI (0.24 vs. 0.17 O.D. *p* < 0.001). In contrast, specific IgG and IgG4 responses to Al-CPI were higher than those to Asc l 5 (*p* < 0.001 in both comparisons, see also Appendix A). In addition, the ratio of the sIgE/IgG4 and sIgE/IgG responses to Asc l 5 were higher than those observed to Al-CPI (Figure 2).

Since it has been hypothesized than concurrent IgG responses may block specific IgE binding activity to allergen/antigens, IgG was depleted from a serum of IgE+, IgG+ individuals, rAl-CPI or rAsc l 5 to test if there were any effects of specific IgE determinations. No significant differences in the O.D. values between samples before or after IgG depletion were found (Appendix A).

### 3.3. The IgE Antibodies to Al-CPI Do Not Elicit an IgE-Mediated Reaction

As shown in Table 1, 41% of subjects living in an ascariasis endemic area had positive a IgE to Al-CPI; however, when allergenic activity was evaluated by different methods, this antibody response did not induce IgE-mediated biological reactions. Ex vivo (HRA or BAT) functional assays were negative in all patients (Figure 3 and Figure 4). In regard to BAT, an *A. lumbricoides* extract at 10 ug/mL induced a CD203c SI higher than two in all cases; however, rAl-CPI did not induce positive results in any of subjects. rAsc l 5 was positive in three out of six patients and had higher mean SI at 10 µg/mL than rAl-CPI 10 µg/mL (Figure 3). 

We confirmed, using histamine release assays, that rAl-CPI did not induce a meaningful response in stimulated basophils; in fact, the percentage of histamine release induced by rAl-CPI was less than 10% in most subjects and concentrations (Figure 4). Histamine release after exposure to rAsc l 5 in sensitized patients has been confirmed in a previous publication [18].

### 3.4. The Antibody Responses to Al-CPI and Asc l 5 Are Not Associated with Asthma

There were no significant differences between asthmatic patients and non-asthmatic control subjects in regard to IgE positivity for either of the two antigens (Table 1). The strength of the antibody responses of the different isotypes was also compared between the two groups, finding no significant differences. Ratios of IgE/IgG4 and IgE/IgG towards rAl-CPI were similar between asthmatic and non-asthmatic subjects. In regard to rAsc l 5, in spite of the fact that we detected its capability to induce an allergic reaction ex vivo [17], specific IgE response was not associated with asthma (Table 1).

### 3.5. Humoral Responses to Asc l 5 and Al-CPI Are Associated with A. lumbricoides Egg Burden

The frequency of infection in the study sample was 67.2% (median egg counts: 1508; interquartile range: 650–4160 epg). Positive responses (IgG+, IgG4+ or IgE+) to Al-CPI or to Asc l 5 were not associated with an *A. lumbricoides* infection (Appendix A). In addition, median O.D. values for all analyzed isotypes against rAsc l 5 or rAl-CPI were similar between infected and non-infected subjects (Appendix A). However, the most severely infected subjects had significantly higher IgG levels for rAl-CPI (Median OD: 0.49) and rAsc l 5 (Median OD: 0.41) compared to the rest of the sample study (rAl-CPI: 0.42 and rAsc l 5: 0.37). Specific IgG to rAl-CPI was marginally associated with severe infection (beta: 0.04, S.E.: 0.02, *p* = 0.04) after adjustment for sex and age. Regarding IgG to rAsc l 5, there was no association with severe infection in a similar multivariate model (beta: 0.03, S.E.: 0.015, *p* = 0.052). We also explored the relationships between egg counts (e.p.g.), antibody-related variables and age (Figure 5), finding direct correlations between specific IgG to rAsc l 5 and egg burden (rho: 0.14, *p* = 0.02) and also with subjects’ age (rho: 0.27, *p* < 0.0001). Age inversely correlated with IgE/IgG ratios to rAl-CPI (rho: −0.22, *p* < 0.001) and to Asc l 5 (rho: −0.131, *p* = 0.025).

Considering the potential confounding effect of age in the relationships between antibody responses and egg counts, we ran multivariate generalized linear models to account for the effect of this covariate (and also for sex). As reported in Table 2, sIgE/IgG4 ratios to rAl-CPI and to rAsc l 5 were inversely associated with egg counts. In addition, sIgE/IgG ratios to both proteins and their association with egg burden were analyzed in a similar model, observing non-significant results. 

### 3.6. rAl-CPI Induces Lower IgE Response and Higher IgG2a Than rAsc l 5 in BALB/c Immunized Mice

Mouse models of experimental immunization have been useful for analyzing the allergenic activity and type of immune response to putative allergen molecules. Ovalbumin (OVA) was used as a positive allergenic control. rAsc l 5 induced the highest levels of IgG1 (Figure 6A), and rAl-CPI induced the highest levels of IgG2a antibodies (Figure 6B). rAsc l 5 induced higher IgE titers than rAl-CPI (Figure 6C). In fact, rAl-CPI specific IgE values in immunized mice were similar to those obtained from mice receiving only PBS (mock group). In regard to the passive cutaneous anaphylaxis, we did not detect Evans dye extravasation after Al-CPI injection (Figure 6D). PCA results for rAsc l 5 have been reported previously [17].

## 4. Discussion

Helminth infections are well-known modifiers of allergic response [29], and most of them have a dual effect on the symptoms of allergic diseases. Depending on several factors, ascariasis can increase asthma symptoms or suppress them [16]. *A. lumbricoides* is a source of immunomodulatory molecules and allergens, both with IgE-binding properties [7,8,18,30]. Al-CPI has anti-inflammatory properties with therapeutic potential and Asc l 5 is the fourth *A. lumbricoides* allergen that we have recently registered at the WHO/IUIS allergen nomenclature sub-committee. Knowing the antibody binding properties of parasite antigens can help us to understand the mechanisms of their influence on allergic diseases like asthma, and also to prevent unwanted allergic reactions induced by parasite-derived immunomodulatory products intended for therapeutical purposes or vaccine candidates [28,31]. In addition, since IgG and IgG4 responses are associated with resistance to helminth infection [32] and protection from allergic reactivity, further basic information can be obtained by analyzing them in natural *A. lumbricoides* exposed populations. 

Our results show a lower IgE response to Al-CPI and higher levels of IgG4 and IgG than Asc l 5. In addition, the frequency of IgG4 seropositivity to Al-CPI was almost double the IgE seropositivity (Table 1) while the frequencies of IgG4 and IgE to Asc l 5 were similar, which was supported by the significantly greater IgE/IgG4 ratios to Asc l 5 compared to Al-CPI. Furthermore, in mice, Al-CPI induced higher levels of IgG2a antibodies and lower levels of IgE. Due to its potential use as an anti-inflammatory agent [7,8], it was in our interest to evaluate if Al-CPI had relevant allergenic activity. Our results showed that Al-CPI does not activate basophils or induce histamine release as evaluated in assays using human blood samples. All these findings mean that, in addition to its anti-inflammatory properties, Al-CPI has low allergenic activity, which suggests that it could be used safely in humans. Other cystatins are allergenic, for example, Ani s 4 from *Anisakis simplex* and Fel d 3, a cystatin-like protein described in cats. The sequence similarity between Al-CPI and these allergens stands at 42% and 30%, respectively, and the likelihood of cross-reactivity is minimal. Ani s 4 has the capability to trigger basophil activation [33], and, to date, it remains the sole allergen identified among helminth-derived CPIs (Allfam 005). Other CPIs originating from helminths, such as Ov17, Av17, Bm-CPI-2, NbCys and Cys-1, are widely acknowledged as immunomodulatory molecules. [12,14,34,35]. In regard to the cystatin from cats, Fel d 3, Hales et al. described the lack of IgE binding of the protein to cat-allergic patients [36] and argued the validity of results from a previous publication that led to its identification because it only performed plaque phage assay for allergenic activity evaluation [37]. In contrast, Asc l 5 again showed positive BAT responses in some sensitized patients. In a previous publication, it was also confirmed that this allergen induced histamine release in sensitized individuals [18]. However, it is also important to emphasize that the presence of IgE antibodies to Asc l 5 do not translate to meaningful allergic reactions since BAT was negative in two out of five IgE+ patients. Although, by several methods, allergenic activity of Asc l 5 is confirmed, it is still an open question if this molecule can induce clinically relevant allergic responses in humans.

It is a matter of increasing research into the involvement of human IgG antibodies in the control mechanisms that have been proposed to explain why only a minority of sensitized people are symptomatic [38]. Some studies have analyzed the effect of naturally occurring IgG and IgG4 responses as a control mechanism in allergic manifestations and also in follow-up studies of allergen specific immunotherapy [38,39,40,41,42]. Santos et al. found that mast cells treated with sera containing IgG4 antibodies from tolerant peanut-sensitized individuals avoid activation induced by IgE from peanut-allergic patients [43]. However, another work found no association between IgG isotypes and any proxy of control mechanisms of allergic diseases such as lower degranulation activity of effector cells (as evaluated using rat basophil leukemia (RBL) assay) [44]. As such, it is possible that the low allergenic activity of Al-CPI is related to its capacity for inducing more IgG4 than other allergenic components in the parasite. Although allergic diseases and helminth infections share several mechanisms of type 2 immunity, including the increase of total and specific IgE, helminths are also able to modulate the inflammation through regulatory cytokines (i.e., IL-10 and TGF-β) leading to the production of IgG4 antibodies [45]. Since Al-CPI induces the production of IL-10 and TGF-β [7], this may partially explain the antibody profile induced during ascariasis. Also, it could be speculated that the previously described inhibition of antigen processing by cystatins could be a possible mechanism of controlling the type of immune response by the MHC. However, it is important to note that although the IgG/IgG4 responses might represent specific blocking antibodies [46,47], they could also be the result of a natural response without blocking activity reflecting immunomodulation [48]. Further studies should define experimentally whether the anti-Al-CPI IgG4 antibodies we observed really have blocking activity. We did not find a significant influence on IgE binding to any of the two antigens due to competition for epitope occupancy by IgG; however, we cannot discard the idea that IgG may have an inhibitory effect of mast cells or basophil degranulation, with both cell types having FcγRIIB receptors that inhibit FcεRI mediated pathways when activated [49]. In addition, another unmet need is to compare the cytokine profile induced by allergenic and non-allergenic *A. lumbricoides* molecules, which would expand the knowledge of mechanisms underlying the antibody pattern we have observed in both humans and mice. 

We found that rAl-CPI and rAsc l 5 have different thermal stabilities as detected by CD. Al-CPI is a thermostable protein as are other members of the CPI protein family of nematodes (including its homologue from *Anisakis simplex*, Ani s 4) [50]. In contrast, the heat-treated rAsc l 5 denatured at 58.2 ± 0.37 °C, which is an important finding because there is no data about thermal stability determined by CD, nor about the constant of the denaturation temperature (tm) from other members of the SXP/RAL-2 protein family. Thermostability is a parameter that is well appreciated in food allergy [51] studies and has been broadly studied for *Anisakis simplex* allergens. For example, the allergen Ani s 4 even resists autoclave treatment (at 121 °C) used in thermal industrial steps of fish processing, retaining its IgE-binding properties and allergic activity [50,52]. Thus, the low allergenic activity of Al-CPI seems to be independent of thermostability and further studies are needed to determine the causes of this interesting property. In addition, here we report for the first time the MS-based sequencing of the natural Al-CPI protein in the *A. lumbricoides* extract. High coverage of the mature sequence was detected. Since other CPIs from nematodes have been described as secreted proteins, our results show that Al-CPI is present in the somatic extract of the parasite and can induce immune responses during natural infection. This cystatin was found to be an abundant protein in the analyzed extract. The influence of abundance on modulation of IgE/IgG responses is not well-defined.

In the studied population (Santa Catalina, Colombia), the prevalence of ascariasis has been between 56 and 62.5% for 10 years, as reported in two separate studies with sample collections in 2004 and 2014 [22,53]. This implies that reinfection cycles occur, leading to an endemic infection. High exposure to *A. lumbricoides* induces immunological responses influenced by the load of infection. Here, we found that specific IgG antibodies to Al-CPI are correlated with egg burden and subject’s age, probably as a result of a prolonged exposure. Interestingly, IgE/IgG4 ratios to rAl-CPI were inversely associated with *A. lumbricoides* egg load after adjustment by age and sex. Similar results were reported by McSharry, C., et al., who found that, in endemic areas, people with significantly more IgE antibodies to ABA-1, an antigen found in the pseudocelomic fluid of Ascaris, which have a lower burden of adult parasites [21]. This suggests that resistance to ascariasis is associated with IgE antibodies to ABA-1. ABA-1 is one of the best characterized allergenic proteins in nematodes [54,55,56], present in both *A. lumbricoides* and *A. suum*. Therefore, the potential effects of the IgE/IgG4 to Al-CPI on resistance to ascariasis deserve further investigations.

Several limitations should be stated. Although their physico-chemical characterization and the evaluation of their biological activity support a good folding, since natural antigens were not isolated and compared to the recombinant counterparts, the obtained results with rAsc l 5 and rAl-CPI may not completely reflect natural exposure. Second, although IgG depletion experiments did not change IgE binding to the recombinant antigens, the effects of this isotype response on degranulation assays were not explored. Third, although rAl-CPI induced low IgE production in mice under routinary protocols for immunization, we cannot rule out the fact that it could take place after longer exposure or in the presence of other parasite components that serve as adjuvants. Our conclusions are limited to the statement that, under the same protocol of immunization, rAl-CPI induced lower IgE production than rAsc l 5 [18] as well as other allergens tested in our lab [57]. Fourth, IgE/IgG ratios may be better estimated from titration curves using standards and further assays should be performed in this regard.

## 5. Conclusions

During ascariasis, helminth-derived components induce different types of humoral immunity. Our findings strongly suggest that the immunomodulatory protein Al-CPI has low allergenic activity, which supports its potential use as an anti-inflammatory product in humans. In contrast, Asc l 5 induces histamine release and the degranulation of cells typically activated by IgE. The clinical impact of this allergenic activity is not clear since sensitization to Asc l 5 is not associated with asthma presentation. Comparing the types of immune responses to parasite antigens is an essential step in understanding the dual effect of helminth infections on allergic diseases.

## Figures and Tables

**Figure 1 biology-12-01340-f001:**
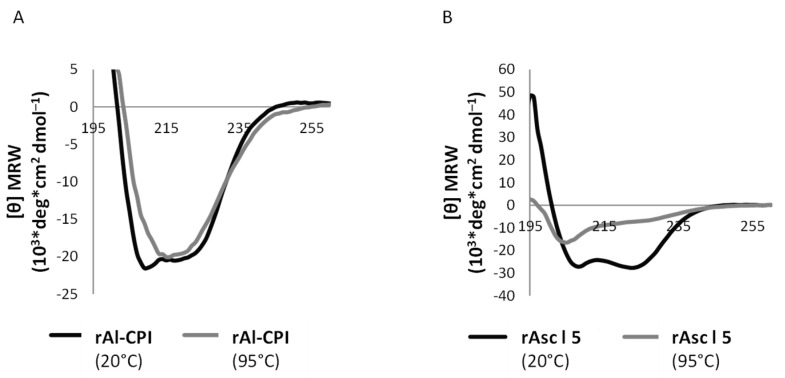
Circular dichroism spectra of rAl-CPI and rAsc l 5. (**A**) rAl-CPI at concentration of 100 µg/mL as detected at 20 and 95 °C. (**B**) rAsc l 5 at concentration of 100 µg/mL as detected at 20 and 95 °C.

**Figure 2 biology-12-01340-f002:**
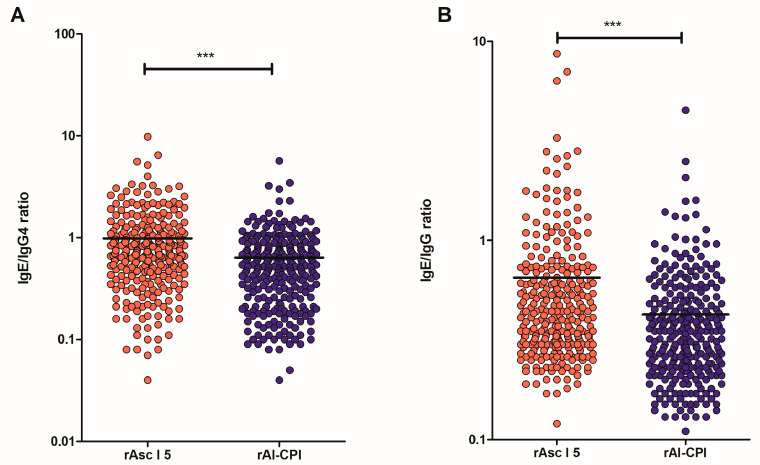
IgE/IgG4 (**A**) and IgE/IgG ratios (**B**) to rAsc l 5 and rAl-CPI. Comparisons were made using Mann–Whitney test. *** *p* < 0.001.

**Figure 3 biology-12-01340-f003:**
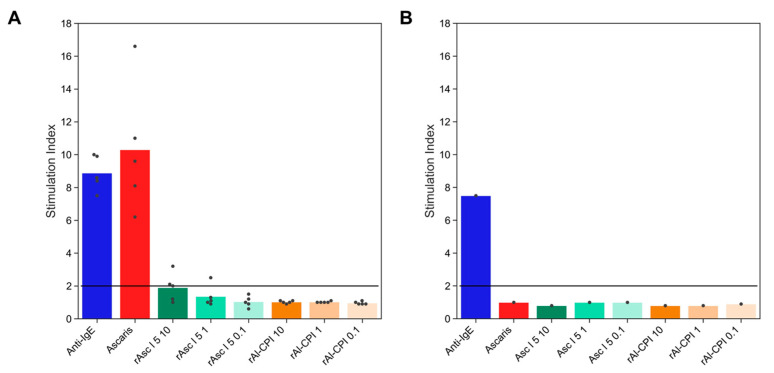
CD203c-based basophil activation test results with Ascaris antigens. Stimulation index (SI) of basophils stimulated with Anti-IgE as positive control, *A. lumbricoides* extract (10 µg/mL), recombinant Asc l 5 (rAsc l 5) and recombinant Al-CPI (rAl-CPI) at different concentrations (0.1, 1 and 10 µg/mL) in rAl-CPI-sensitized subjects. (**A**), bars correspond to mean values of SI in bi-sensitized patients (*n* = 5). Dots represent individual data. In (**B**), bars show SI values for a negative control subject who was non-sensitized to *A. lumbricoides* neither rAl-CPI. Horizontal lines indicate the threshold to consider a positive BAT test.

**Figure 4 biology-12-01340-f004:**
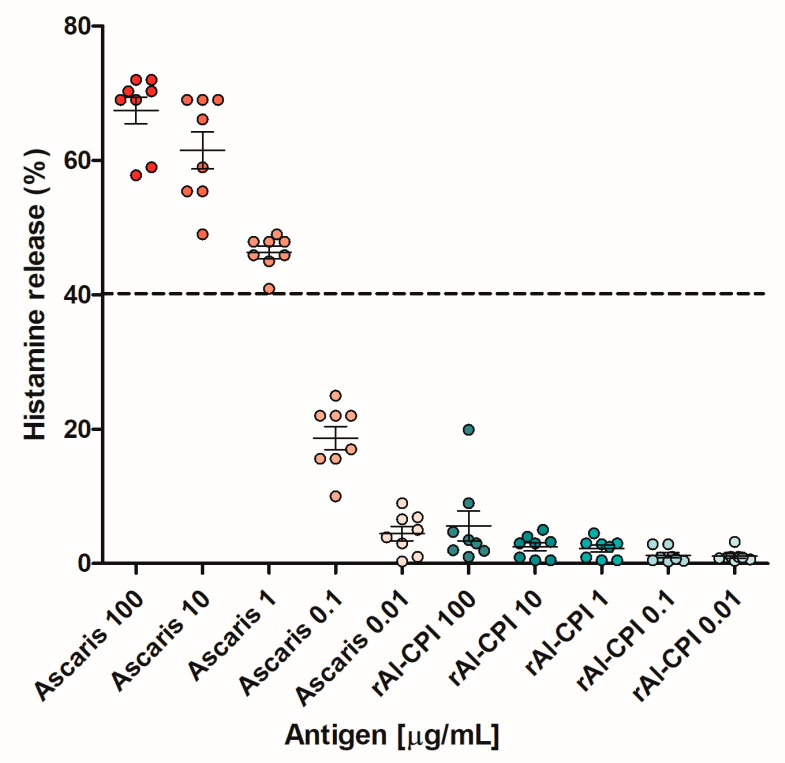
Histamine release induced by *A. lumbricoides* extract and recombinant Al-CPI (rAl-CPI). Fresh blood from eight patients was stimulated with different concentrations of extract or rAl-CPI. Each dot represents the result for one patient. Bar errors represent the mean ± standard error of the mean (SEM) of OD values for each condition. The dotted line represents the cut-off value to define a histamine release test as positive (40%).

**Figure 5 biology-12-01340-f005:**
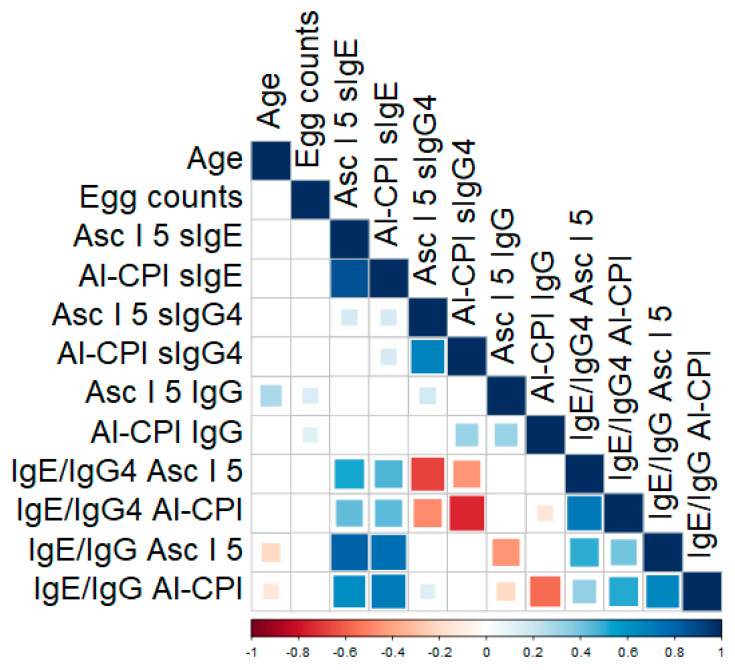
Correlation of humoral responses with *A. lumbricoides* egg burden and age. Only significant correlations are shown as colored squares. The red–blue scale indicates Spearman coefficient (Rho) values from −1 to 1.

**Figure 6 biology-12-01340-f006:**
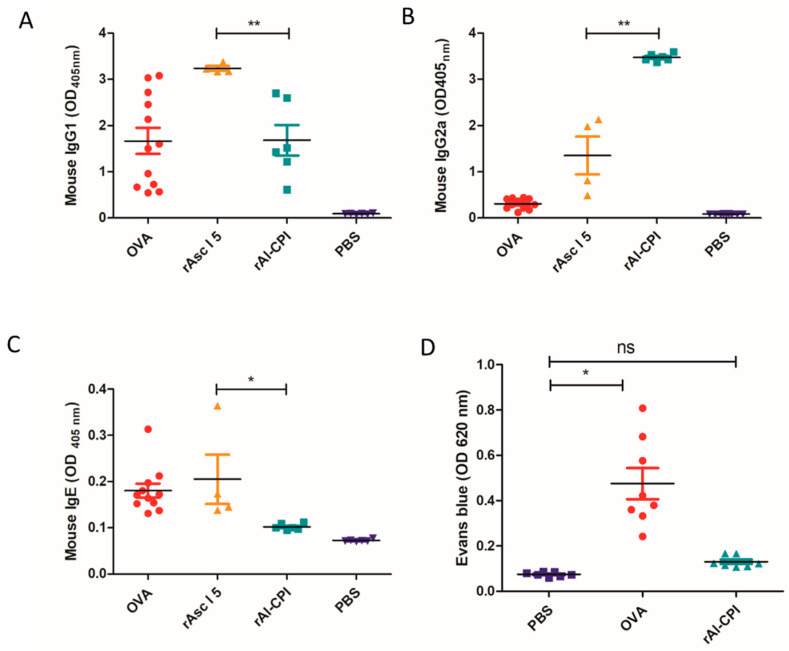
Analysis of antibody responses in BALB/c mice. (**A**) Specific −IgG1 (**B**) –IgG2 and (**C**) −E values to OVA, rAsc l 5, rAl-CPI and PBS. (**D**) Extracted Evans blue after PCA reaction detected to 620 nm. Mean ± standard error of the mean (SEM) of OD values are shown. Comparison between groups was performed via one-way analysis of variance (ANOVA) and Dunnett’s multiple comparison test: * *p* < 0.05, ** *p* < 0.01, (ns) no significance for *p* > 0.05. OD: Optical density units as detected by ELISA. OVA (Ovalbumin).

**Table 1 biology-12-01340-t001:** Relationship between antibody responses to Al-CPI and Asc l 5 with asthma.

	Total Sample Size [*N* = 293]	Non-Asthmatics [*n* = 246]	Asthmatics [*n* = 47]	*p*-Value
Age in years [mean ± SD] *	30.9 ± 21.5	31.8 ±21.2	26.4 ± 22.3	0.13
Sex, female n (%) **	208 (71)	171 (69.5)	37 (78.7)	0.2
Asc l 5 [n (%)] **				
IgE+	153 (52.2%)	125 (50.8%)	28 (59.6%)	0.27
IgG4+	201 (68.6%)	165 (67.1%)	36 (76.6%)	0.18
IgG+	206 (70.3%)	173 (70.3%)	33 (70.2%)	0.99
Al-CPI [n (%)] **				
IgE+	120 (41.0%)	97 (39.4%)	23 (48.9%)	0.23
IgG4+	237 (80.9%)	200 (81.3%)	37 (78.7%)	0.68
IgG+	222 (75.8%)	184 (74.8%)	38 (80.9%)	0.38

* Mann–Whitney test was used for continuous variable comparisons and ** χ^2^ test for proportions.

**Table 2 biology-12-01340-t002:** Association of Ascaris egg counts in feces and IgE/IgG4 ratios to rAl-CPI and rAsc l 5.

Variable	β	S.E.	*p*-Value
Al-CPI			
IgE/IgG4_ratio	−0.24	0.12	0.046 *
Gender	0.18	0.13	0.16
Age (years)	0.05	0.002	0.48
Asc l 5			
IgE/IgG4_ratio	−0.08	0.07	0.28
Gender	0.20	0.13	0.13
Age (years)	0.002	0.003	0.54

* Significant, *p* < 0.05; S.E.: standard error.

## Data Availability

Research experimental data from subjects and animals can be requested to the corresponding author by e-mail.

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
