# Peer review of "Comparison of Antibody Responses against Two Molecules from Ascaris lumbricoides: The Allergen Asc l 5 and the Immunomodulatory Protein Al-CPI"

_biology, 2023, doi:10.3390/biology12101340_

Round 1

Reviewer 1 Report

In this maniscript, Ahumada, V. et al. evaluate the antibody response against two molecules from Ascaris and show differences in the allergenicity and allergenic activity for both, together with differences in the IgG response. The results were discussed in the context of the potential immunomodulatory properties of the pathogen. The work is pretty simple, did not required complex experimental techniques but still very relevant for the scientific community. I only have some minor comments:

- The authors state that the two molecules that they use for all the experiments "induce" specific antibodies in humans. The capacity to induce a response is true and demonstrated in mice, but for humans, I do not agree that the appropriate word here is "induce". For the evaluation it is presumed that humans produced antibodies against natural molecules (not the recombinants) and those antibodies also bind to the recombinants.

-Why was rAsc l 5 not used in the evaluation of IgE-mediated reactions? This paper is dealing with a discussion where the comparison between the antibody response and biological properties of rAsc l 5 and Al-CPI is necessary.

- It would be appropriate to comment what is known about the natural allergens Al-CPI and rAsc l 5, and how similar the recombinants are to them. Using the recombinants for these sort of experiments and analysis should be supported by comparartive studies of natural vs recombinant, to avoid bias in the conclusions.

-Line 57. The authors say that sensitization to Asc l 5 WAS 52% in the general population. What's the current situation? Still 52% or maybe higher?

- Please be careful with the grammar and writting. Some sections are not well written or small things like making sure that all scientific names are in italic (line 49 for example). The english can be improved to make the manuscript excellent.

-The MS analysis says that the Ascaris extracts contains Al-CPI, and that's it. I do not think it has something to do with the availability. This should be corrected.

-Line 361-362: Please use references to support the statements. 

-I undersand well the discussion about the allergenic activity of Al-CPI, but, in the context of the HRA and BAT of this study, how does the IgG influence the observations presented?

- Line 217: Delete "almost" 

Please make sure to write all scientific names properly. And re-check the whole manuscript because some sentences could not be so easy to read. But generally speking, the quality of english language is high

Author Response

In this manuscript, Ahumada, V. et al. evaluate the antibody response against two molecules from Ascaris and show differences in the allergenicity and allergenic activity for both, together with differences in the IgG response. The results were discussed in the context of the potential immunomodulatory properties of the pathogen. The work is pretty simple, did not required complex experimental techniques but still very relevant for the scientific community. I only have some minor comments:

- The authors state that the two molecules that they use for all the experiments "induce" specific antibodies in humans. The capacity to induce a response is true and demonstrated in mice, but for humans, I do not agree that the appropriate word here is "induce". For the evaluation it is presumed that humans produced antibodies against natural molecules (not the recombinants) and those antibodies also bind to the recombinants.

Answer: We definitely agree with the reviewer in regard of the use of “induce” term. We have changed along the manuscript all terms that misuse “induce” when we refer to the use of recombinant proteins for measuring antibody in humans. In fact, we also change the manuscript title as follows:

“Comparison of antibody responses against two molecules from Ascaris lumbricoides: the allergen Asc l 5 and the immunomodulatory protein Al-CPI”

-Why was rAsc l 5 not used in the evaluation of IgE-mediated reactions? This paper is dealing with a discussion where the comparison between the antibody response and biological properties of rAsc l 5 and Al-CPI is necessary.

Answer: The reason for this was that evaluation of rAsc l 5 allergenicity was already published in ref. 18; however, as we did have simultaneous evaluation of BAT for Asc l 5 and Al-CPI in five patients, we prepare a new figure with all data together.

- It would be appropriate to comment what is known about the natural allergens Al-CPI and rAsc l 5, and how similar the recombinants are to them. Using the recombinants for these sort of experiments and analysis should be supported by comparative studies of natural vs recombinant, to avoid bias in the conclusions.

Answer

There are not studies about the fold of natural Al-CPI por Asc l 5; however, the circular dichroism analysis in this investigation supports we have produced a well-folded recombinant that is also compatible with the information derived from the RMN-solved structure published by Mei et al (014) PLoS One 9: e96069-e96069). We also found that rAl-CPI immunized mice could detect natural CPI in the extracts. However, as we did not isolate native Al-CPI and perform direct comparisons of both counterparts (native and recombinants) we will add this limitation to the discussion.

Line 475,“Several limitations should be stated. Although their physico-chemical characterization and the evaluation of their biological activity support a good folding, native antigens were not isolated and compared to the recombinant counterparts; then, the obtained results with rAsc l 5 and rAl-CPI might not completely reflect natural exposure…”

-Line 57. The authors say that sensitization to Asc l 5 was 52% in the general population. What's the current situation? Still 52% or maybe higher?

Answer: we referred to the sample study. We have changed this sentence to: “The IgE response to Asc l 5 was about 52% in the sample study”

- Please be careful with the grammar and writting. Some sections are not well written or small things like making sure that all scientific names are in italic (line 49 for example). The english can be improved to make the manuscript excellent.

Answer: Thanks for your comment. We have carefully checked all document and revise grammar and typos.

-The MS analysis says that the Ascaris extracts contains Al-CPI, and that's it. I do not think it has something to do with the availability. This should be corrected.

Answer: It is hypothesized that availability may influence allergenicity by several authors. We improved redaction and organized paragraph for a better understanding. We also added a new experiment to show the recognition of Al-CPI by a pAb antisera from Al-CPI immunized mice.

Results (Line 255)

We also assessed by Western Blot if anti-Al-CPI antibodies raised in immunized mice were able to recognize the natural antigen in A. lumbricoides extract, finding two bands in the 10-13 kDa range, which is in accordance with the expected size for Al-CPI (Supplementary Figure 1). Al-CPI content in the extract was also confirmed by tandem mass spectrometry (MS/MS) sequencing. High sequence coverage (85.7%) of the mature protein (Al-CPI GenBank accession number: HQ404231.1) was obtained (Supplementary Figure 2). Based on proteomic analysis of the extract, Al-CPI was the top 5th most abundant protein in the preparation; in contrast, Asc l 5 had lower representativeness in the extract.

-Line 361-362: Please use references to support the statements. 

Answer:

We added two previously cited reference above to support the statements about Al-CPI anti-inflammatory activity:

“All these findings mean that, in addition to its anti-inflammatory properties (7,8), rAl-CPI has low allergenic activity, which suggests that it could be used safely in hu-mans”

  1. Coronado S, Barrios L, Zakzuk J, Regino R, Ahumada V, Franco L, et al. A recombinant cystatin from Ascaris lumbricoides attenuates inflammation of DSS-induced colitis. Parasite Immunol. 2017;39(4).
  2. Coronado S, Zakzuk J, Regino R, Ahumada V, Benedetti I, Angelina A, et al. Ascaris lumbricoides Cystatin Prevents Development of Allergic Airway Inflammation in a Mouse Model. Front Immunol. 2019;10:2280.

-I undersand well the discussion about the allergenic activity of Al-CPI, but, in the context of the HRA and BAT of this study, how does the IgG influence the observations presented?

Answer:

We have added new data using IgG depletion with protein G to test its influence on IgE binding:

In results:

“Since it has been hypothesized than concurrent IgG responses may block specific IgE binding activity to allergen/antigens, IgG was depleted from serum of IgE+ IgG+ individuals rAl-CPI or rAsc l 5 to test if there was any effect of specific IgE determinations. There was not found any significant differences in the O.D. values between samples before or after IgG depletion (Supplementary Figure 4)”

In Discussion, we added new lines related with this topic:

Line 429-437: However, it is important to note that although the IgG/IgG4 responses might represent specific blocking antibodies [46, 47], they also could be the result of a natural response without blocking activity reflecting immunomodulation [48]. Further studies should define experimentally if the anti-rAl-CPI IgG4 antibodies we observed really have blocking activity. We did not find a significant influence on IgE binding to any of the two antigens due to competition for epitope occupancy by IgG; however, we cannot discard that IgG may have an inhibitory effect of mast cells or basophil degranulation, both cell types, having FcγRIIB receptors that inhibit FcεRI mediated pathways when activated [49].

- Line 217: Delete "almost" 

Answer: The word “almost” was deleted. “the number of subjects with IgG4+ responses to rAl-CPI was twice as high as the frequency rate”

Comments on the Quality of English Language

Please make sure to write all scientific names properly. And re-check the whole manuscript because some sentences could not be so easy to read. But generally speking, the quality of english language is high

Answer: Thanks. We have carefully checked English writing and corrected typos.

Reviewer 2 Report

In this manuscript, Ahumada et al. compare IgE, IgG, and IgG4 antibody responses to two A. lumbricoides proteins (Asc l 5 and Al-CPI), previously characterized as a putative allergen or with anti-inflammatory properties, respectively. Specific responses have been tested against two recombinant proteins, rAsc I 5 and rAl-CPI, using sera obtained from a Colombian population where A. lumbricoides infestation is endemic. Additionally, these proteins have been tested as immunogens in mice.

The results seem to indicate that there could be a different antibody response in both humans and immunized mice against the aforementioned proteins. Although the study is potentially interesting, there are flaws that need to be adressed to support the author's conclusion.

Major comments:

  • As the overall aim of the study is to demonstrate lower allergenicity of rAl-CPI compared to its allergenic counterpart (rAsc l 5), it is surprising that such a comparison is lacking for functional tests (histamine release and basophil activation) aimed at demonstrating that IgE antibodies reacting with rAl-CPI are not functional. Given that IgE antibodies raised against both proteins in humans are quite similar (sTable 3), the positive control using an A. lumbricoides extract is not sufficient, and results with rASc I 5 should be presented to support the authors’ conclusion.
  • The authors remark in the Results and Discussion sections the higher levels of IgG and IgG4 against rAl-CPI compared to rAsc I 5, but no blocking studies have been performed. As the authors themselves state, these studies would be necessary. It is not clear for the reviewer why then they did not conduct such relatively simple in vitro but important assays while including other complex assays that are less necessary in the context of the study (see below)
  • Some headings in the Results section are confusing as they mix results that disrupt the reading flow on the main topic of the manuscript. In this regard, the justification for the thermostability experiments (circular dichroism) or the identification of Al-CPI (mass chromatography) in the A. lumbricoides extract is not clearly explained and appears out of context. Please clarify this issue and reorder the Results section accordingly.

Minor comments:

  • The heading of the Result section "rAl CPI induces lower IgE specific antibodies than rAsc l 5 but higher IgG/IgG4 responses" is misleading, as the recombinant proteins are not immunogens and therefore cannot induce any response. Please change to Al CPI and Asc l 5.
  • Please include an Ethics Statement for blood sample collection.
  • Please correct typing errors, e.g., "rection" instead of "reaction" (line 134).

Author Response

In this manuscript, Ahumada et al. compare IgE, IgG, and IgG4 antibody responses to two A. lumbricoides proteins (Asc l 5 and Al-CPI), previously characterized as a putative allergen or with anti-inflammatory properties, respectively. Specific responses have been tested against two recombinant proteins, rAsc I 5 and rAl-CPI, using sera obtained from a Colombian population where A. lumbricoides infestation is endemic. Additionally, these proteins have been tested as immunogens in mice.

The results seem to indicate that there could be a different antibody response in both humans and immunized mice against the aforementioned proteins. Although the study is potentially interesting, there are flaws that need to be addressed to support the author's conclusion.

Major comments:

  • As the overall aim of the study is to demonstrate lower allergenicity of rAl-CPI compared to its allergenic counterpart (rAsc l 5), it is surprising that such a comparison is lacking for functional tests (histamine release and basophil activation) aimed at demonstrating that IgE antibodies reacting with rAl-CPI are not functional. Given that IgE antibodies raised against both proteins in humans are quite similar (sTable 3), the positive control using an A. lumbricoides extract is not sufficient, and results with rASc I 5 should be presented to support the authors’ conclusion.

Answer: Dear reviewer. Thanks for all comments. The reason for the lack of comparisons with rAsc l 5 allergenicity was that it is already published in ref. 18. This paper mainly focuses on the allergenic characterization of Al-CPI, an immunomodulatory protein of Ascaris, However, as we did in fact had simultaneous evaluations of Asc l 5 and Al-CPI by BAT in 5 patients, we prepared a new figure including columns for rAsc l 5.

Since the kits for histamine release assays are difficult to find for us, we did not repeat experiments for rAsc l 5. In ref 18, there are reports of the ability of this recombinant protein to elicit histamine release in exposed basophils from fresh blood.

  • The authors remark in the Results and Discussion sections the higher levels of IgG and IgG4 against rAl-CPI compared to rAsc I 5, but no blocking studies have been performed. As the authors themselves state, these studies would be necessary. It is not clear for the reviewer why then they did not conduct such relatively simple in vitro but important assays while including other complex assays that are less necessary in the context of the study (see below)

Answer:

We have performed experiments of that kind and we now have added these data to the article:

“Since it has been hypothesized than concurrent IgG responses may block specific IgE binding activity to allergen/antigens, IgG was depleted from serum of IgE+ IgG+ individuals rAl-CPI or rAsc l 5 to test if there was any effect of specific IgE determinations. There was not found any significant differences in the O.D. values between samples before or after IgG depletion (Supplementary Figure 4)”.

  • Some headings in the Results section are confusing as they mix results that disrupt the reading flow on the main topic of the manuscript. In this regard, the justification for the thermostability experiments (circular dichroism) or the identification of Al-CPI (mass chromatography) in the A. lumbricoides extract is not clearly explained and appears out of context. Please clarify this issue and reorder the Results section accordingly.

Answer: You are right. We have moved results from circular dichroism to the beginning of Results section and have also changed the title of this sub-section to “Circular dichroism reveals recombinant Asc l 5 and Al-CPI are well-folded products that differ in their thermostability”. We consider this information is important to support we have obtained good recombinant products to analyze the response against antigens from Ascaris to which humans in endemic areas are naturally exposed. The capacity of recombinant Al-CPI immunized mice to recognize the native counterpart protein also support that its immunogenicity is the retained in the recombinant we produced. Similar results were shown in a previous publication of our group  for Asc l 5 (ref 18).

Minor comments:

  • The heading of the Result section "rAl CPI induces lower IgE specific antibodies than rAsc l 5 but higher IgG/IgG4 responses" is misleading, as the recombinant proteins are not immunogens and therefore cannot induce any response. Please change to Al CPI and Asc l 5.

Answer: We changed the title of this section as follows:

“Lower IgE levels but higher IgG/IgG4 response was found for Al-CPI than for Asc l 5”

  • Please include an Ethics Statement for blood sample collection.

Answer:

This information is included at the end of the manuscript:

Informed Consent Statement: A full verbal explanation of the investigation was given to each participant and written informed consent was obtained from all subjects or their parents or legal guardians for participation in the study, including blood sample collection.

  • Please correct typing errors, e.g., "rection" instead of "reaction" (line 134).

Answer: Thanks. We have checked all manuscripts for more typos and further corrections.

Reviewer 3 Report

The paper describes some interesting results in comparing the antibody responses to two different Ascaris proteins and the possible pharmacological activity of  the cystatin designated Al-CPI

The median/mean bar in Figure 1 does not show up at all well especially for the rAl-CPI with the blue dots. Recommended that the bar be made more obvious or perhaps put arrows on the side.

The paper does not describe gravimetric estimates of the antibody titres  which would provide more definitive results and put some perspective on the study with respect to size of the responses and the strength of the proposed immunomodulation. It is suggested that this added to the "to do list" in the discussion

It is assumed that the IgE/IgG4 ratio was calculated from the O.D. values. This should be added to the methods. The ratio should at least be calculated from titrations curves against standards and as per the titres be better calculated with gravimetric estimates. At least add this to the to do list.

The comparison in figure 3 of basophil degranulation would best include Asp l 5 in keeping with the theme of the paper.

The study made structural analyses of the recombinant proteins but failed to present them with respect to the important issue of the structural integrity needed for the serology. Clearly the results from table 1 could just results from the efficacy of detection due to folding and plating efficiency. Comment on this

Results on IgE and IgG titres instead of frequencies and isotype ratios would provide more convincing important data but presumably these did not show the desired results.

The experiments on immunising mice are very interesting. The first point to note is that very sub-optimal regimen has been used with respect to IgE induction and this might help the detection of the low responses to to Al-CPI (rapid "boosting" decreases not increases responses). It is not expected that this can be corrected for the current submission but what can be included is the PCA titre for the Asc l 5 or adding it to the "to do list" in the discussion.

An obvious experiment is to mix the Asc l 5 and Al-CPI in mouse immunisation (but not needed for the current "special issue" submission)

Although it is too involved to consider in detail in the current submission the nature of the allergenicity of cystatins is far from resolved. Ichikawa (doi: 10.1046/j.1365-2222.2001.01169.x.) and later Hales (doi: 10.1016/j.jaip.2013.08.008.) failed to detect IgE antibody to purified rFel d 3 and Roesner (doi: 10.18176/jiaci.0737) only found it in atopic dermatitis patients associated with an auto-immune response to human cystatin

Author Response

Comments and Suggestions for Authors

The paper describes some interesting results in comparing the antibody responses to two different Ascaris proteins and the possible pharmacological activity of  the cystatin designated Al-CPI

The median/mean bar in Figure 1 does not show up at all well especially for the rAl-CPI with the blue dots. Recommended that the bar be made more obvious or perhaps put arrows on the side.

Answer: Thanks for your comments. They certainly have improved the quality of the manuscript. According to your first suggestion, size of mean lines have been increased and put on top of dots for a better visualization.

The paper does not describe gravimetric estimates of the antibody titres  which would provide more definitive results and put some perspective on the study with respect to size of the responses and the strength of the proposed immunomodulation. It is suggested that this added to the "to do list" in the discussion

It is assumed that the IgE/IgG4 ratio was calculated from the O.D. values. This should be added to the methods. The ratio should at least be calculated from titrations curves against standards and as per the titres be better calculated with gravimetric estimates. At least add this to the to do list.

Answer:

This sentence was added to Methods to explain how IgE/IgG or IgE/IgG4 ratios were calculated. “IgE/IgG and IgE/IgG4 ratios were calculated from the end-point OD values for the three isotypes using IgE as the numerator.”

Also, we added to the Limitations list the lack of standards for an optimal estimation of antibody ratios:

Line 486: “Fourth, IgE/IgG ratios may be better estimated from titration curves using standards and further assays should be performed in this regard.”

The comparison in figure 3 of basophil degranulation would best include Asp l 5 in keeping with the theme of the paper.

The reason for the lack of comparisons with rAsc l 5 allergenicity was that it is already published in ref. 18. This paper mainly focuses on the allergenic characterization of Al-CPI, an immunomodulatory protein of Ascaris, However, as we did in fact had simultaneous evaluations of Asc l 5 and Al-CPI by BAT in five patients, we prepared a new figure including columns for rAsc l 5.

The study made structural analyses of the recombinant proteins but failed to present them with respect to the important issue of the structural integrity needed for the serology. Clearly the results from table 1 could just results from the efficacy of detection due to folding and plating efficiency.

Answers: We agree with you that this is a limitation of the study because we did not purify native Al-CPI or Asc l 5 to fully compare the recombinant products with their counterpart; however, we did perform a detailed structural characterization of recombinant proteins using circular dichroism  and biological activity evaluation. The rAl-CPI we produced  is an active protease inhibitor . The activity of metal-binding protein of Asc l 5 was also confirmed. These findings reduce the uncertainty of having obtained mis-folded recombinant products. However, we added as a limitation of the study a sentence related with your comment:

Several limitations should be stated. Although their physico-chemical characterization and the evaluation of their biological activity support a good folding, since native antigens were not isolated and compared to the recombinant counterparts; then, the obtained results with rAsc l 5 and rAl-CPI might not completely reflect natural exposure.

Comment on this

Results on IgE and IgG titres instead of frequencies and isotype ratios would provide more convincing important data but presumably these did not show the desired results.

Results of IgG also support our findings, supplementary Figure 3 contains these results.

We added these sentences to Reuslts:

Median O.D. values for sIgE determinations to rAsc l 5 were greater than those against Al-CPI (0.24 vs 0.17 O.D. p<0.001). In contrast, specific IgG and IgG4 responses to rAl-CPI were higher than to rAsc l 5 (p<0.001 in both comparisons, see also Supplementary Figure 3).

The experiments on immunising mice are very interesting. The first point to note is that very sub-optimal regimen has been used with respect to IgE induction and this might help the detection of the low responses to to Al-CPI (rapid "boosting" decreases not increases responses). It is not expected that this can be corrected for the current submission but what can be included is the PCA titre for the Asc l 5 or adding it to the "to do list" in the discussion.

Answer: We have a similar opinion to that stated in your comment. It is probable that under a longer protocol of immunization, rAl-CPI can induce higher IgE responses; however, what we can state is that using the same mice strain, adjuvant, antigen concentration and protocol for immunization, rAsc l 5 induces higher IgE production than Al-CPI. We have added this limitation to the related paragraph:

Third, although rAl-CPI induce low IgE production in mice under routinary protocols for immunization, we cannot rule out that it could take place after longer exposure or in the presence of other parasite components that serve as adjuvants. Our conclusions are limited to state that under the same protocol of immunization, rAl-CPI induce lower IgE production than rAsc l 5 [18] as well as other allergens tested in our lab [56].

An obvious experiment is to mix the Asc l 5 and Al-CPI in mouse immunisation (but not needed for the current "special issue" submission)

 Answer: thanks for your suggestion. It is an interesting experiment.

Although it is too involved to consider in detail in the current submission the nature of the allergenicity of cystatins is far from resolved. Ichikawa (doi: 10.1046/j.1365-2222.2001.01169.x.) and later Hales (doi: 10.1016/j.jaip.2013.08.008.) failed to detect IgE antibody to purified rFel d 3 and Roesner (doi: 10.18176/jiaci.0737) only found it in atopic dermatitis patients associated with an auto-immune response to human cystatin

Answer: thanks for these suggestions. We added two of these references to Discussion.

Lines 404-408 “In regard to the cystatin from cat, Fel d 3, Hales et al described the lack of IgE binding of the protein to cat allergic patients [36] and argued the validity of results from a previous publication that led to its identification because it only performed plaque phage assay for allergenic activity evaluation [37]”.

Reviewer 4 Report

I would like to thank all the authors for their valuable work. They tried to investigate the sensitization pattern and immune response to two  Ascaris lumbricoides anigens in an attempt to provide new insights on allergen immunotherapy. I have only few points to consider

Line 13. There is no need for simple summary in presence of abstract.

Line 50 please clarify the abbreviation DSS

Line 201 please correct Ig4 to IgG4

Line 380 please correct T2 to T helper 2 immunity

Line 419 please calrify the abbreviation ABA-

1

Minor English editing and revision are required

Author Response

Comments and Suggestions for Authors

I would like to thank all the authors for their valuable work. They tried to investigate the sensitization pattern and immune response to two  Ascaris lumbricoides antigens in an attempt to provide new insights on allergen immunotherapy. I have only few points to consider

Answer: Thanks for reading carefully the manuscript and for all your valuable comments.

Line 13. There is no need for simple summary in presence of abstract.

Answer: This is a suggestion of Journal Guidelines; that is why we have two abstracts, one with technical language and the other for any type of reader.

Line 50 please clarify the abbreviation DSS

Answer: We have replaced the abbreviation for the full name of the detergent: “Dextran sodium sulphate”

Line 201 please correct Ig4 to IgG4

Answer: Corrected

Line 380 please correct T2 to T helper 2 immunity

Answer: We changed to type 2 immunity which is a broader term that also include innate immune cells that polarize T helper responses to Th2.

Line 419 please calrify the abbreviation ABA-

1

ABA-1 is not an abbreviature, it is a given name to an Ascaris lumbricoides antigen. It has been named by the first investigator that isolate it from Ascaris extracts.

 McGibbon AM, Christie JF, Kennedy MW, Lee TD. Identification of the major Ascaris allergen and its purification to homogeneity by high-performance liquid chromatography. Mol Biochem Parasitol. 1990 Mar;39(2):163-71. doi: 10.1016/0166-6851(90)90055-q. PMID: 1690856.

Round 2

Reviewer 2 Report

The manuscript has improved significantly with respect to its first version. However, there are some aspects that still need to be considered.

Major

As I can see from the new Figure 3, the differences between rAl-CPI and rAsc l 5 are marginal in terms of basophil activation, certainly below the cut-off established for the assay in both cases (by the way, the cut-off dotted line should be maintained in the figure as in the previous version). On the other hand, also in both cases, the induced histamine production is generally low (Figure 4 for rAl-CPI and reference 18 for rAsc l 5). Consequently, a quite different behavior of these proteins as allergens, as stated by the authors, should be considered with caution and, therefore, some emphatic phrases pointing out these differences should be rephrased. For example, the last sentence of the Simple Abstract, and others in the same sense throughout the manuscript including the Conclusion.

 Minor

- Lines 32-33. Skin prick tests are mentioned in the abstract as if they were an outcome, but these tests have not been shown or discussed in this manuscript. Please change accordingly.

- Line 202. Please clarify the phrase "mouse pAb rabbit IgG from mice sensitized with rAl-CPI and PBS." What does "pAb" mean? What is the function of rabbit IgG here? Please also explain in the corresponding figure legend.

- Line 206. Please clarify why mention “rabbit IgG”.

- Lines 254-257. It would be clarifying for the reader if reference were made here to different Pm observed between the native and recombinant protein (Supplementary Figure 1).

- Lines 280-281. P values are indicated in the text, but not in Supplementary Figure 3. Please mark the statistical significance with the corresponding arsteristic(s) in this figure.

- Line 291. Please clarify what Al-16 means in Supplementary Figure 4, and why the specific IgE values in O.D. are so different (much higher) than those shown in Supplementary Table 3. By the way, Supplementary Figure 4 is missing the letters A and B describing each graph.

- Line 297/307/309. Change Al-CPI to rAl-CPI

- Line 310. Change Asc l 5 to rAsc l 5

- Line 323. Change Asc l 5 to rAsc l 5 and Al-CPI to rAl-CPI

- Line 330 and the rest of this paragraph: Change Asc l 5 to rAsc l 5 and Al-CPI to rAl-CPI

- Line 349/363: Change Al-CPI to rAl-CPI

- Line 392/394/395/397: Change Al-CPI to rAl-CPI

- Lines 403-407: Please rewrite this sentence for clarity.

- Line 407. Change Asc l 5 to rAsc l 5

- Line 442. Change Asc l 5 to rAsc l 5 and Al-CPI to rAl-CPI

- Line 445. Change recombinant Asc l 5 to rAsc l 5

-    - Line 451. Change Al-CPI to rAl-CPI

Author Response

The manuscript has improved significantly with respect to its first version. However, there are some aspects that still need to be considered.

Major

As I can see from the new Figure 3, the differences between rAl-CPI and rAsc l 5 are marginal in terms of basophil activation, certainly below the cut-off established for the assay in both cases (by the way, the cut-off dotted line should be maintained in the figure as in the previous version). On the other hand, also in both cases, the induced histamine production is generally low (Figure 4 for rAl-CPI and reference 18 for rAsc l 5). Consequently, a quite different behavior of these proteins as allergens, as stated by the authors, should be considered with caution and, therefore, some emphatic phrases pointing out these differences should be rephrased. For example, the last sentence of the Simple Abstract, and others in the same sense throughout the manuscript including the Conclusion.

Answer:

We have replaced the figure 3 for another graph that displays individual SI values as dots inside bars (depicting mean values). It is true that rAsc l 5 did not induce basophil activation in all patients, but as it is observed by individual cases, in some others it did induce SI above the threshold (Line has been added again). This is different to what is observed with Al-CPI. None of the 5 subjects had a positive test to this molecule.

In regard to histamine release, what we found in the last paper of Asc l 5 was up to 40% of histamine release in one Asc l 5 sensitized patient and insignificant histamine release in the negative control subject. No more subjects were tested.

Considering your comments, we have included the following changes:

In simple summary:

Line 20“Our results show that both molecules induce specific antibodies, but in contrast to rAl-CPI, rAsc l 5 activate cells associated with allergic reactions in some individuals. All together, these data suggest that these molecules have differences in the elicited immune response.”

In Discussion:

Line 410: “However, it is also important to emphasize that the presence of IgE antibodies to Asc l 5 do not translate to meaningful allergic reactions since BAT was negative in 2 out of 5 IgE+ patients. Although, by several methods, allergenic activity of Asc l 5 is confirmed, it is still an open question if this molecule may induce clinically relevant allergic responses in humans.”

Line 496: “Our findings strongly suggests that the immunomodulatory protein Al-CPI has low al-lergenic activity, which supports its potential use as an anti-inflammatory product in humans. In contrast, Asc l 5 induces histamine release and degranulation of cells typically activated by IgE. The clinical impact of this allergenic activity is not clear since sensitization to Asc l 5 is not associated with asthma presentation.”

 Minor

- Lines 32-33. Skin prick tests are mentioned in the abstract as if they were an outcome, but these tests have not been shown or discussed in this manuscript. Please change accordingly.

Answer: We apologize for the mistake. Skin prick test was removed from the abstract.

- Line 202. Please clarify the phrase "mouse pAb rabbit IgG from mice sensitized with rAl-CPI and PBS." What does "pAb" mean? What is the function of rabbit IgG here? Please also explain in the corresponding figure legend.

- Line 206. Please clarify why mention “rabbit IgG”.

Answer: We apologize for the mistake. We used a similar protocol for Western Blot using Rabbit or Mouse polyclonal IgG; then we used the paragraph from the English version of our protocol and erroneously did not erase the word “rabbit” for this specific experiments.

Now in methods:

Mouse IgG from rAl-CPI sensitized mice and PBS-injected mouse (mock) were diluted 1:1000 in blocking buffer and incubated for one hour at RT with A. lumbricoides extract or rAl-CPI transferred to PVDF membranes after SDS-PAGE separation. Strips were rinsed twice and washed three times more with 0.1% Tween 20 PBS. Horseradish peroxidase conjugated anti mouse- IgG (diluted 1:100.000 in blocking buffer) was used as secondary antibody and incubated for 1 hour at RT. Membrane was rinsed twice and washed three times more with 0.1% Tween 20 PBS. Chemiluminiscence reaction was developed by addition of 1:1 mixture of the SuperSignal West Femto Maximum Sensitivity Substrate™ and incubation for 5 minutes in the dark. Images were obtained at different times in a specialized chemilumiscence image detector (G: Box, Syngene, UK).

And in supplementary figure legend 1:

Supplementary Figure 1 Recognition of CPI in the natural extract of Ascaris lumbricoides. Left,  Anti-Al-CPI polyclonal antibody raised from three rAl-CPI immunized mice (pAb-Al-CPI)was tested against rAl-CPI (Lane 1) or the natural extract of A. lumbricoides (Lane 2). Right, same experiments were performed with PBS-immunized mice.

- Lines 254-257. It would be clarifying for the reader if reference were made here to different Pm observed between the native and recombinant protein (Supplementary Figure 1).

Answer: This line was added to the legend of Supplementary Figure 1:

Difference in the molecular weight (MW) between native and recombinant Al-CPI is due to the extra amino acids added in the recombinant plasmid.

- Lines 280-281. P values are indicated in the text, but not in Supplementary Figure 3. Please mark the statistical significance with the corresponding arsteristic(s) in this figure.

Answer: Sup. Figure 3 has been updated, adding p-values. Aesthetics of the graphs was also improved.

- Line 291. Please clarify what Al-16 means in Supplementary Figure 4, and why the specific IgE values in O.D. are so different (much higher) than those shown in Supplementary Table 3. By the way, Supplementary Figure 4 is missing the letters A and B describing each graph.

Answer: Al16 corresponds to Asc l 5. Al16 is the old name we used to have for Asc l 5 due to its similarity with As16, an antigen from Ascaris suum. Letters A and B were added to the figure. As you can observe in Supplementary Figure 3, there are sera (although few) with high O.D. to rAl-CPI or rAsc l 5. Supplementary Table 3 display median O.D. levels.

- Line 297/307/309. Change Al-CPI to rAl-CPI

- Line 310. Change Asc l 5 to rAsc l 5

- Line 323. Change Asc l 5 to rAsc l 5 and Al-CPI to rAl-CPI

- Line 330 and the rest of this paragraph: Change Asc l 5 to rAsc l 5 and Al-CPI to rAl-CPI

- Line 349/363: Change Al-CPI to rAl-CPI

- Line 392/394/395/397: Change Al-CPI to rAl-CPI

Answers: All changes have been done. Thanks.

- Lines 403-407: Please rewrite this sentence for clarity.

Answer:

“The sequence similarity between Al-CPI and these allergens stands at 42% and 30%, respectively, and the likelihood of cross-reactivity is minimal. Ani s 4 has the capability to trigger basophil activation [33], and to date, it remains the sole allergen identified among helminth-derived CPIs (Allfam 005). Other CPIs originating from helminths, such as Ov17, Av17, Bm-CPI-2, NbCys, and Cys-1, are widely acknowledged as immunomodulatory molecules”

- Line 407. Change Asc l 5 to rAsc l 5

- Line 442. Change Asc l 5 to rAsc l 5 and Al-CPI to rAl-CPI

- Line 445. Change recombinant Asc l 5 to rAsc l 5

-    - Line 451. Change Al-CPI to rAl-CPI

Answers: All changes have been done. Thanks.